# A Qualitative Investigation of the Acceptability and Feasibility of a Urinary Tract Infection Patient Information Leaflet for Older Adults and Their Carers

**DOI:** 10.3390/antibiotics10010083

**Published:** 2021-01-16

**Authors:** Leah F. Jones, Heidi Williamson, Petronella Downing, Donna M. Lecky, Diana Harcourt, Cliodna McNulty

**Affiliations:** 1Public Health England, Gloucester GL1 1DQ, UK; leah.jones@phe.gov.uk (L.F.J.); pdowning16@gmail.com (P.D.); donna.lecky@phe.gov.uk (D.M.L.); 2Health and Social Sciences, Frenchay Campus, University of the West of England, Bristol BS16 1QY, UK; Heidi3.williamson@uwe.ac.uk (H.W.); Diana2.Harcourt@uwe.ac.uk (D.H.)

**Keywords:** urinary tract infections (UTI), older adults, qualitative, leaflet, Theoretical Domains Framework, antimicrobial resistance, antibiotics

## Abstract

Urinary tract infections (UTIs) can be life threatening in older adults. The aim of this study was to primarily understand the acceptability and feasibility of using a UTI leaflet for older adults in care homes and the community. Qualitative interviews and focus groups informed by the Theoretical Domains Framework were conducted in 2019 with 93 participants from two English areas where a UTI leaflet for older adults had been introduced to improve self-care advice. Discussions were conducted with care staff (carers and nurses), older adults, general practice staff (GPs, nurses and health care assistants), and other relevant stakeholders and covered experiences of using the leaflet; its implementation; and barriers and facilitators to use. Participants deemed the leaflet an acceptable tool. Clinicians and care staff believed that having information in writing would reinforce their messages to older adults. Care staff reported that some older adults may find the information overwhelming. Where implemented, care staff used the leaflet as an educational guide. Clinicians requested the leaflet in electronic and paper formats to suit preferences. Implementation barriers included lack of awareness of the leaflet, lack of staffing and resource, and weak working relationships between care homes and general practices. It is recommended that regional strategies must include plans for dissemination to care homes, training, promotion and easy access to the leaflet. Improvements to the leaflet consisted of inclusion of antibiotic course length, D-mannose, atrophic vaginitis and replacement of less alarmist terminology such as ‘life threatening’.

## 1. Introduction

Urinary tract infections (UTIs) are one of the most common causes of hospitalisation in care home residents, posing a significant threat to life in this age group [1]. Most UTIs are caused by the bacterium *Escherichia coli* (*E. coli*). *E. coli* bloodstream infections (BSIs) rates in the UK have increased by 33.8% since 2012/2013 [2], with the highest rates of *E. coli* bacteraemia observed amongst older adults over the age of 75 [3,4].

The focus of any suspected infection is often difficult to determine in older adults, especially if they have dementia, and therefore clinicians may use point-of-care tests to help determine the focus of infection or use empirical broad-spectrum antibiotics. Despite strong evidence to suggest that antimicrobial therapy to treat asymptomatic bacteriuria (ASB) is unnecessary and potentially harmful [5], urine dipsticks are often used by primary care and care home staff [6]. As a result, a proportion of older adults are misdiagnosed with UTIs rather than ASB and may receive unnecessary antibiotics [7,8].

Combined with improved diagnostic pathways for health care workers, information leaflets can help explain to older adults in the community and in care, clinicians’ diagnostic decisions and management plans, providing self-care and prevention advice to patients to improve patients’ understanding, and self-care skills. Leaflets can be highly valued by patients and can help reduce unnecessary antibiotic prescribing [9,10,11]. In 2017, Public Health England (PHE) developed an evidence-based UTI leaflet for older adults and their carers (community and care home carers) to increase patients’ and carers’ knowledge about ASB, UTIs and antimicrobial resistance (AMR) and increase their skills to recognise, help prevent and self-care for UTIs. The pictorial leaflet can be found in Appendix A, [12] and provides an anatomical illustration of the urinary system, prevention information, signs and symptoms of UTI, other causes of confusion, self-care advice, what to expect from a clinical consultation, a section on AMR, and safety netting for pyelonephritis and sepsis.

The aim of this study was to:Explore the acceptability and feasibility of using the leaflet.Understand the perceived value of the leaflet.Identify barriers and facilitators to using the leaflet.Understand how the leaflet interacts with UTI diagnostic tools and other resources.Inform further developments to the leaflet.Explore potential indications of behaviour change.

The Theoretical Domains Framework (TDF) [13] is a behavioural model designed to help understand implementation. The TDF was used in the development phase of the leaflet by informing the interview schedules and was used in this study to structure the interview schedules to ensure all behavioural determinants were explored including knowledge, skills, and environmental context.

## 2. Results

Ninety-three participants took part in either focus groups or interviews from March to September 2019, from a range of urban and rural locations across Gloucestershire and East Kent. For a detailed figure of the recruitment figures and strategy, please see Appendix A

Of the 93 participants, 53 were carers working in care homes (3 nurses, 2 administrators, and 48 carers), 4 care home residents, 25 general practice staff servicing care homes and the community (13 GPs, 10 nurses and 2 health care assistants), and 8 stakeholders. Stakeholders included representatives from the Royal College of General Practitioners (RCGP; the professional body for general practitioners in the United Kingdom), the National Health Service Improvement (NHSI; responsible for overseeing foundation trusts and NHS trusts, as well as independent providers that provide NHS-funded care), the Care Association Alliance (CAA; a membership association for local Care Associations to exchange best practice), an academic pharmacist, and members of four Clinical Commissioning Groups (CCGs, clinically-led statutory NHS bodies responsible for the planning and commissioning of health care services for their local area), in East Kent (3) and Gloucestershire (1).

### 2.1. Key Findings

#### 2.1.1. The Acceptability and Feasibility of Leaflet Use in Primary Care and Care Home Settings

All participants reported that the leaflet is a suitable tool for care homes and general practice and that they would like the leaflet to be available in both electronic and hard-copy formats. Suggestions for dissemination included giving the leaflet to residents’ families and friends as well as to the residents themselves; displaying the leaflet as a waiting room resource; giving the leaflet to patients during consultations for suspected UTI and providing it at the reception desk or next to urine submission boxes when urine samples are submitted. Most older adults would be happy to receive the leaflet, although some concern was raised by clinicians as to the leaflet’s acceptability for older adults who are coping with multiple health issues and who may find the information overwhelming.

#### 2.1.2. Value of the Leaflet

All participants valued the leaflet for various reasons, including
-Written information reinforces their advice to older adults (clinicians and care staff),-It is an educational guide for care staff (care staff) and friends and family (older adults), and-It has flexibility for use in other infection prevention and control (IPC) areas (commissioners), with other age groups (clinicians and older adults).

Some participants suggested that the leaflet would be suitable for community pharmacy and out-of-hours (OOH) settings, although one nurse practitioner believed that implementation in OOH settings would be very difficult due to transient staff.

#### 2.1.3. Barriers and Facilitators

Despite reporting local dissemination efforts through the provision of the leaflet, local champions and local hydration campaigns, lack of awareness by GP staff and care staff was the biggest barrier to leaflet use. Although some clinicians believed it was their role to cascade information to care homes, most in this study did not, and therefore weak working relationships between care homes and general practices could contribute to lack of implementation, although this will vary across regions and between facilities. Commissioning teams reported that high turnover of care staff, lack of resources and staffing issues in the CCG meant they are unable to visit every general practice and care home to promote the leaflet and conduct training around UTI diagnosis and management.

In one region, the IPC lead reported utilising ‘links practitioners’ in every general practice and care facility to promote and disseminate their training and resources, but also believed further work is needed to establish whether this approach is effective, as one link nurse reported attending training but not feeding back about materials.

#### 2.1.4. Comments on the Leaflet

Research findings are represented in Table 1. However, many of the leaflet findings do not directly fit into the TDF domains [14] and are therefore represented below with quotes.

Participants suggested minor improvements to the leaflet content including the use of less alarmist terminology. One participant said *“It might get people panicking about life-threatening…we currently have two residents with full capacity and the one would be straight on the phone.”* Care home staff 3.

Others suggested inclusion of the NICE recommendation of three-day antibiotic courses for proven UTI to address patient expectations for longer antibiotic courses, inclusion of the NICE recommendation of D-mannose (a type of sugar) dietary supplement as a self-care preventative option in recurrent UTI, and mention of atrophic vaginitis as an alternative cause of urinary symptoms in post-menopausal women.

Staff sharing the leaflet liked all sections, reporting that they reinforced the information that they gave to patients. One stated: *“I find them helpful if I’m having a discussion with a patient and they’re not really buying into what I’m saying…it’s a little bit of extra evidence that I’m not some weird doctor trying to make up stuff.”* General practice staff 3.

A CAA representative recommended increasing leaflet dissemination via the CAA to ensure delivery directly to care homes. The RCGP representative recommended a short RCGP screencast to raise awareness amongst general practitioners (GPs).

#### 2.1.5. Leaflet Interaction with UTI Diagnostic Tools and Other Resources

GP staff reported that they mostly used the leaflet alongside either PHE or locally developed UTI diagnostic tools as part of quality improvement initiatives within general practice. Some reported that quality improvement was their overall goal.

One stakeholder was optimistic that their work in implementing UTI training, leaflet use and diagnostic guidance had reduced the number of unrequested urines being submitted by patients to several GP practices in their region.

#### 2.1.6. Indications of Behaviour Change Following Use or Implementation of the Leaflet

All commissioners intend to continue their promotion of the leaflet alongside the diagnostic flow charts within their local UTI or infection prevention campaigns.

Care staff are motivated by wanting the best for their residents and improving their wellbeing, and they reported consistent use of genital hygiene and hydration strategies as prevention methods with all residents. Barriers, reported by older adults and care staff, to implementing these prevention strategies for older adults included reluctance to drink in order to avoid regular toilet visits and limited incontinence pad allowance, although care homes also reported buying additional pads to supplement their allocations. One stated: *“because I keep going at night. Which isn’t right…I’m not drinking more. I hopefully am drinking less.”* Care home resident 2

Some care staff intended to cease use of urine dipsticks moving forwards. However, some care staff intended to continue using them and a few GPs were using urine culture to inform diagnostic decision making; dipsticking and culture were perpetuated by unrequested urine samples being dropped off at general practice receptions and perceived pressure from care staff to use urine dipsticks. One stakeholder had reduced unrequested urine submissions in several GP practices in their region through the combined use of training and implementation of the leaflet and diagnostic guidance.

Care staff reported being confident in their ability to identify changes in resident behaviour which indicated illness, but had some difficulty in distinguishing UTIs from other illnesses due to similar presentations. However, despite the leaflet providing signs and symptoms of UTI alongside other causes of confusion, clinicians reported that they had difficulty in diagnosing UTI in patients with dementia. Most clinicians reported that diagnosis of UTI was complicated by atypical presentations, vague symptoms reported by carers, incontinence and other conditions presenting like a UTI. Therefore, clinicians requested further information and resource to support diagnosis in this group.

#### 2.1.7. Key Findings and the Theoretical Domains Framework

To identify important behavioural determinants and using a deductive approach, key themes were placed into the corresponding 14 domains of the TDF listed in Box 1. The first key theme, ‘Use and implementation of the UTI leaflet’, addresses objectives 1–5 in order to determine the leaflet’s acceptability, the feasibility of its use, perceived value, interaction with other resources, barriers to using the leaflet and potential further developments to the leaflet. The second and third key themes(Identifying/diagnosing a UTI’ and ‘Managing and/or treating a UTI) address objective 6, which explores potential indications of behaviour change by examining current behaviours around UTI diagnosis and management. This process is illustrated in Table 1, with each of the three themes used as column headings.

Box 1The domains of the Theoretical Domains Framework.
**The 14 domains of the Theoretical Domains Framework**
1. Knowledge2. Skills3. Social/professional role and identity4. Beliefs about capabilities5. Optimism6. Beliefs about Consequences7. Reinforcement8. Intentions9. Goals10. Memory, attention and decision processes11. Environmental context and resources12. Social influences13. Emotion14. Behavioural regulation

## 3. Discussion

### 3.1. Summary

All participants including care staff, general practice staff, older adults and stakeholders reported that the leaflet is a valuable IPC tool, suitable for care homes and general practice, that can be used alongside diagnostic tools and antibiotic guidance, reinforcing messages to older adults while also providing a useful guide for care staff. Participants believed that younger adults would benefit from the leaflet and it should be provided in other health settings.

Participants provided valuable suggestions for dissemination, such as a provision to families and friends, and placement in clinical waiting rooms, reception areas and next to urine submission boxes.

Lack of awareness of the leaflet was the biggest barrier to its use, and implementation barriers prohibited commissioners from effective dissemination. Suggested changes to the leaflet included use of less alarmist terminology, the inclusion of three-day antibiotic courses, the inclusion of D-mannose, and mention of atrophic vaginitis.

Commissioners reported that they would continue to promote the leaflet locally and some care staff would cease use of urine dipsticks moving forwards. Barriers to preventing UTI which are not easily addressed by the leaflet include difficulties in diagnosing UTI in older adults with dementia, and reluctance by older adults to hydrate sufficiently to help reduce their toilet visits.

### 3.2. Comparison with Existing Literature

Patient information leaflets on antibiotic prescribing for a variety of conditions have led to reductions in antibiotic use [15,16]. However, process evaluation information, such as the acceptability and feasibility of the patient information leaflets, is rarely reported. In a systematic review of leaflet effectiveness [15], many studies only report clinical outcomes, and are therefore limited by not assessing patient acceptability or feasibility of use in real-world settings outside of controlled trial conditions.

To the best of our knowledge, there have not been any leaflets developed for older adults on the topic of UTI, and therefore drawing comparisons across different leaflets, audiences and conditions was inherently difficult. A qualitative study evaluating an interactive information booklet for parents of young children with respiratory symptoms ‘When should I worry?’ explored the views and opinions of parents and clinicians as part of a trial measuring the booklet’s effectiveness [17]. Francis et al. found that parents and clinicians valued the leaflet and many parents had kept the booklet for future references. They concluded that the role of leaflets and other information resources can help facilitate effective communication, and indeed, the link between effective communication of health information with clinical outcomes is well documented [18]. Despite some similarities to the present study, these findings must be accepted with caution as leaflets can vary in content and quality, and therefore perceived value will vary.

A qualitative study exploring patient views of medication information leaflets found that leaflet font size, paper quality, writing style and size of the paper are important factors for enhancing readability [19]. We addressed these design parameters iteratively during the development of the UTI leaflet, which may explain why there were few criticisms relating to its aesthetics [20].

A similar study by Fleming, et al. [21] explored antibiotic prescribing in long-term care facilities using the TDF and Behaviour Change Wheel (BCW), and recommended the provision of education on the topic of antibiotics, prescribing guidelines and AMR, with the provision of management guidelines and supporting evidence. Even though Fleming et al. [21] explored general antibiotic prescribing across conditions without a UTI focus, a similar recommendation from this study included the provision of education for care homes on UTI diagnosis, management and ASB. The current study further supports the need for additional resources in care homes, specifically around UTI education and the provision of guidance on ASB, diagnosis of UTI and urine dipstick use. Using urine dipsticks to diagnose UTI in older adults is not accurate due to high rates of ASB [22,23,24]. GP staff reported using the leaflet as part of quality improvement programmes. The leaflet compliments the To Dip or Not To Dip (TDONTD) [8] quality improvement programmes that aim to improve the diagnosis and management of UTI in older patients. Regional evaluations of TDONTD have found significant reductions in UTI antibiotics prescribed in care home residents, unplanned hospital admissions, urosepsis and acute kidney injury [8].

### 3.3. Strengths and Limitations

This is the first study to explore the acceptability and feasibility of a patient-facing evidence-based UTI leaflet for older adults and their carers in the community, including in care home settings. This study offers valuable insight into implementation and attitudes towards the leaflet as well as current diagnostic and management practices across both settings. To the best of our knowledge, in the UK, there is currently no nationally available patient-facing information leaflets for older adults on the topic of UTI, and there are no evaluation studies assessing acceptability and feasibility, or implementation of patient-facing resources on UTIs for any age group.

The present study included a large number of participants from a range of backgrounds. However, care staff and general practice staff may have had greater interest in UTIs, leading to some selection bias. Selection bias was reduced by inviting care homes and GP practices from two regions, approaching potential participants in random order [25] and by providing an incentive to participate. Our findings indicate a wide variation in management and use of the tools, which indicates that selection bias was minimised. However, many participants had not used or seen the leaflet before and were viewing it for the first time as part of this study. This suggests that either implementation strategies by the CCGs could be improved, or that more time was needed between initial leaflet implementation and participant recruitment [26].

A further sample limitation is that only adults with full capacity were approached to take part, and therefore only older adults who could read and understand the leaflet took part, and no data were captured for those older adults deemed unsuitable to receive the leaflet due to the content being ‘overwhelming’. Insights from this group may have proved useful for informing improvements to the leaflet or informing the development of a new leaflet.

Qualitative methodology was employed to gain detailed insight into leaflet use and how it contributes to identifying, managing and preventing UTIs. Using both interviews and focus groups facilitated recruitment as participants could choose the format to suit their preference, and use of both provided breadth of exploration across many individuals, and in-depth exploration with individual experiences and attitudes [27].

### 3.4. Implications

As the older adult UTI leaflet was reported as “invaluable” by patients and carers, GP staff and CCGs reported that they would continue to implement the leaflet “as a guide for patients and carers to help identify and manage UTIs”. We suggest that it should be made available in both electronic and hard-copy formats to suit users’ preferences, as part of a quality improvement program to advance the management of UTIs. However, due to reported implementation barriers, commissioners may want to consider electronic dissemination as an inexpensive and potentially easier method of promotion to care homes and general practices. This could include use of QR codes or integration into GP clinical systems for ease of access including use of computer prompts as reminders.

Following minor changes to terminology, and the inclusion of information about a three-day antibiotic course, D-mannose and vaginal atrophy, the leaflet will correspond to the current PHE UTI diagnostic flowcharts and NICE/PHE UTI guidance information [28,29] and should therefore be disseminated in care homes as an educational guide to staff. The leaflets should also be disseminated to older adult care home residents of any age. However, where residents lack capacity or may find the leaflet overwhelming, the leaflet could be given to families and friends of residents to provide education and to reinforce health behaviour messages from staff around hydration, self-care and prevention.

General practices should consider the provision of the leaflet as a waiting room resource or to be given/emailed to patients during or following consultations, or to be given at reception to educate patients bringing in urine samples. Primary care clinicians using the PHE diagnostic flow charts may also want to consider the leaflet as a complementary triaging resource to reinforce and communicate their diagnostic decisions with patients.

As weak working relationships between care homes and general practices could be a contributing factor to lack of implementation, commissioners should consider promoting the leaflet during training sessions for both care homes and general practices as an infection prevention and control resource. Regional strategies must include plans for widespread dissemination to care homes including monitoring of attendees and non-attenders to training sessions, monitoring of leaflet use with TARGET UTI audits. A greater implementation may be needed in OOH and community pharmacy settings.

National promotional strategies through the RCGP and CAA should be considered to ensure national dissemination. Health Education England has a short video explaining the value of the leaflet and diagnostic flowchart [30].

Currently, there is a separate non-pictorial UTI leaflet for younger adults that has been used in GP and pharmacy settings [31]. As participants reported that the information in the older adult leaflet is relevant to people of all ages and some patients did not relate to the label ‘older adults’, a combined leaflet may be useful. A combined leaflet has been developed, but further work is needed to evaluate this in primary care settings.

## 4. Materials and Methods

Study design: This is a cross-sectional qualitative study using interviews and focus groups informed by the TDF [13].

Leaflet implementation: To understand implementation and usage in a real-world setting, PHE researchers were not involved in the implementation of the leaflet. As such, those unfamiliar with the leaflet were still eligible to participate to understand their management of UTIs, their reasons for not having seen/used the leaflet, and their assessment of the leaflet’s value. All participants were sent the leaflet alongside the study information form to allow reflection prior to the discussions.

Gloucestershire CCGs’ plan for implementing the UTI leaflet included posting the leaflet to all general practices; workshops and educational training on the UTI guidance offered to all care homes and GP practices; a hydration campaign in care homes, promoted using merchandise and a touring marketing bus. The East Kent CCG disseminated the leaflet to all practices and care homes electronically or in hard copy depending on preference. Links practitioners were established in every care home and general practice in East Kent, who were then offered training on the UTI guidance with the view of disseminating it in their respective settings.

Data collection started after the CCGs had implemented the leaflet in each region for a minimum of four months. Each region aimed to saturate their regions with the leaflet but did not monitor uptake to determine whether this had occurred.

### 4.1. Participant Selection and Eligibility

General practice staff and care staff were invited from two CCG regions in the UK, Gloucestershire CCG and East Kent CCG, which were selected due to their intentions to disseminate the leaflet and willingness to support the study. To avoid recruiting only AMR enthusiasts, regions were selected based on antibiotic prescribing at a primary care level [32], and individual facility prescribing data were not explored. Region size and regional demographic variation were taken into account as these can impact implementation and health literacy [33]. Stratification by rural/urban allowed for a variance in participant demographics.

Lists were formed of all care homes and general practices in each region. All care homes and general practices were contacted with an introductory letter describing the study, and the vast majority of facilities did not respond to the initial letter—only a minority responded expressing an interest. Each list was then randomised using Excel’s RAND function, and two weeks following receipt of the letter, care homes and general practices were contacted in random order with a follow-up telephone call.

In accordance with the Enabling Research In Care Homes (ENRICH) guidelines, care homes considered ‘inadequate’ in the CQC inspection rating were not selected for this study [34]. This only equated to 3 care homes across both regions.

Managers or the point of contact were asked to disseminate the study information to recruit staff and older adults. Managers/contacts were requested to approach older adults with experience of UTI and who were able to provide informed consent.

Stakeholders were identified using known contacts through PHE and previous engagement with professional societies, with the aim to recruit national representatives of primary care clinicians and care staff, as well as commissioners of primary care services to discuss their implementation and regional strategies.

All participants gave written and verbal consent and were offered £20 in vouchers; staff were offered certificates of participation as well as vouchers.

### 4.2. Data Collection

General practice staff and care staff were offered interviews or focus groups depending on their preference. Focus groups were conducted in a quiet room provided by the facility and were heterogenous i.e., all job roles were permitted to attend. Interviews were either face to face or via telephone, depending on participant preference. Older adults were offered interviews rather than focus groups, as discussing experiences of UTI could be considered personal. However, three older adults from one care home requested a focus group.

Seventeen interviews were conducted and lasted 13–47 min, and 12 focus groups containing between 3 and 10 individuals lasted for 24–57 min. After each, discussion field notes were made of important topics and non-verbal data. All interviews and focus groups were conducted by one researcher (LJ).

### 4.3. Interview Schedules

Questions were informed by the TDF [13] and the qualitative findings from the needs assessment to develop the leaflet [6].

The schedules were semi-structured and used flexibly (see Appendix A). Interviews and focus groups with general practice staff and care staff covered their leaflet use, and barriers and facilitators to usage. Discussions with older adults explored their experiences of having UTIs, their attitudes and opinions of receiving the leaflet, its content and its perceived usefulness. Interviews with stakeholders focused on organisational barriers or facilitators to implementation. The interview schedules were piloted with 1–2 people from each group, and pilot data were included in the results as no major amendments were made.

The care home managers viewed the older adult interview schedule in order to ensure they were aware of the questions being asked to their residents and identified older adults with sufficient understanding to participate. However, managers were asked to keep schedules confidential to prevent potential priming.

### 4.4. Data Analysis

Transcripts were analysed by one researcher (LJ) in Nvivo 11 [35] using Inductive Thematic Analysis (ITA). Following ITA, a deductive approach was adopted by placing key themes into the domains of the TDF to identify important behavioural determinants. A double coder (PD) coded 10% (3) of the transcripts.

### 4.5. Researcher Context

The primary investigator, LJ, has previous experience of using the TDF, conducting research in this area with care staff, GP staff, and older adults as part of the leaflet development work. Researcher bias in this study has been mitigated by utilising patient input into the interview schedule development, use of a double coder and by presenting the results to both regions and receiving their feedback.

## 5. Conclusions

This novel study has provided insights into the acceptability and feasibility of using the UTI leaflet for older adults and their carers in general practice and care home settings, including current diagnostic and management practices, variation in implementation, and barriers and facilitators. Consequently, this study highlights the ways in which the leaflet has influenced recognition and treatment behaviours, and also ways to improve the leaflet, implications for successful implementation, and suggestions for ways in which new interventions could overcome the barriers to appropriate UTI diagnosis and management. A combination of new complementary interventions, and improvements to the leaflet and its implementation will be needed in order to further influence behaviour change in this context.

## Figures and Tables

**Table 1 antibiotics-10-00083-t001:** Key findings and corresponding TDF domains covering use and implementation of the UTI leaflet, UTI diagnosis/identification and UTI management in older adults.

Use and Implementation of the UTI Leaflet	Identifying/Diagnosing a UTI	Managing and/or Treating a UTI
**Awareness** The majority of care staff had not seen the leaflet before. *“I can’t even say that I’ve seen them in the waiting rooms or anything.”* Care home staff 1 ***(Knowledge)*** **Leaflet content** The majority of care staff believed that residents will not understand the content of the leaflet. *“if you want the residents to read, this is too much for them.”* Care home staff 7 ***(Beliefs about consequences)*** Most older adults did not like the title ‘older adults’ as they do not associate themselves with the label. *“the only thing I didn’t like about it was the wording at the top which says it’s a leaflet for older adults and carers.”* Older adult 2 ***(Professional role and identity)*** Stakeholders stated that because the leaflet links with hydration they can link it to many areas of infection prevention such as respiratory infections and AMR. *“I think at a time when people are feeling the pinch, they’re very happy for messages that crossed over several goals, really.”* Stakeholder 6 ***(Reinforcement)*** **Implementation** One OOH practitioner felt that the leaflet would be very difficult to implement in OOH settings. *“so I* work *in out of hours as well and the, it’s not something that I routinely translate across into out of hours…there are certain things that you have to follow when you do out of hours work.”* Nurse practitioner 3 ***(Environmental context and resources)*** CCG stakeholders reported that high turnover of care staff makes implementation difficult. *“It was a two day course and it’s like painting the Forth Bridge, due to the turnover. Somebody said to me, what about the rest of (location) and I said, that’s a full time job.”* Stakeholder 3 ***(Environmental context and resources)*** All CCG stakeholders stated that they did not have enough resource to provide education to all care homes and GP practices. *“we’ve got so many care homes I haven’t got enough time in the day, as well as 70 odd GP practices.”* Stakeholder 4 ***(Environmental context and resources)*** Most general practice staff did not believe it is their role to cascade information to care homes. *“if you go to the care homes and you do in care homes one by one it will work very well…Rather than you doing with the GP practice and then you think GP practice will influence the care homes.”* General practice staff 2 ***(Professional role and identity)*** One stakeholder suggested that difficulties in implementation in OOH is due to transient staff. *“The people who run out of hours say to me, anything that’s implemented nationally or best practice, in out of hours is probably 12, 18 months later. Because they work with a bit of a more transient locum population”* Stakeholder 3 ***(Social influence)*** All GP staff expressed the intention to implement or use the leaflet. *“I will print it off and I will give, …I definitely will because I do like giving people information …so yeah, that is definitely something I will use.”* Nurse practitioner 1 ***(Intentions)*** All CCG stakeholders intended to continue their implementation work of the leaflet and wider complimentary resources. *“next year…we’re planning to run a day to really train people in* how *to improve their practice…that’s how I really hope to roll it out.”* Stakeholder 2 ***(Intentions)*** Commissioner stakeholders stated that they have no way of monitoring leaflet use. *“I’ve got no way of knowing whether they used those leaflets.”* Stakeholder 3 ***(Behavioural regulation)*** **Leaflet use** Care staff that had used the leaflet used it as their guide for identifying and managing UTIs. *“It’s our guide for how we appoint* (identify) *this UTI.”* Care home staff 7 ***(Environmental context and resources)*** The general practice staff using the leaflet tended to also use PHE’s national diagnostic and treatment guidelines, or their own adapted version of the guideline as a complementary resource. *“We’ve all got, the flowcharts we’ve got them all in colour, they’re laminated, they’re in all the rooms.”* Nurse practitioner 2 ***(Environmental context and resources)*** One practitioner would not use the leaflet with the over 85s as they feel it could be too much for some. *“it’s knowing your patient well enough to think, is this going to add to my consultation or actually are we just better off talking very, very simply and having that as a conversation…rather than saying here’s some information which backs up what we’ve talked about. I would spend more time with that older patient so that they feel more comfortable in knowing that information.”* Nurse practitioner 3 ***(Memory, attention and decision processes)*** Two older adults passed the leaflet on to friends and family. *“What I’ve done is, I’ve photocopied yours…just to give to my daughters* because *this sort of information is invaluable.”* Older adult 1 ***(Intentions)*** **Accessibility** General practice staff would like the leaflet to be made available electronically and in hard copy to suit their preferences for dissemination, as some prefer texting or emailing leaflets whereas others prefer providing hard copies. *“So, bits of paper get lost in piles but if you’ve got it electronically, so you can print it off or text it to them, it’s easier.”* GP 2, telephone interview ***(Environmental context and resources)*** **Attitudes and intentions** All care staff believed the leaflet will be a useful tool to help staff and relatives identify and manage UTIs. *“I think that might help the* relatives *understand a little bit more.”* Care home staff 3 ***(Environmental context and resources)*** All older adults believed the leaflet would help with the identification and management of UTIs better. *“I read the leaflet and yes, it’s very helpful…when I looked at the worsening signs of urine infection I’ve had all those when it’s been at its worst and I think people should know what it is and what to expect.”* Older adult 3 ***(Beliefs about consequences)*** Most older adults felt that the leaflet would benefit younger adults too. *“it’s not just for older people, is it? I mean it’s for, a lot of young people get it as well. So why is it targeted to* older *people?”* Older adult 2 ***(Beliefs about consequences)*** One stakeholder believed that the leaflet would reduce the demand for antibiotics. *“**one thing that I kept hearing was about GPs feeling pressured by patients for antibiotics.* So, *what I think …. it will really impact on how health professionals manage and therefore then that will have a knock on.”* Stakeholder 1 ***(Beliefs about consequences)*** Some general practice staff reported that their overall goal was quality improvement. *“the thing is quality* improvement…*there’s no point in doing stuff if you’re not actually making a difference or it’s going to be useful to you.”* General practice staff 1 ***(Goals)*** A few general practice staff wanted to use the leaflet to educate those bringing in urine samples to reception. *“to have at reception actually…for the people that don’t get as far as the waiting room* and *they drop in a sample or want to drop in a sample.”* General practice staff 3 ***(Intentions)*** As detailed in ‘beliefs about consequences’, all older adults were optimistic that the leaflet could have a positive effect on UTI management, but care staff were pessimistic about the utility of the leaflet with many older adults. ***(Optimism)***	**Urine dipsticks** A minority of care staff were aware that they should not be using dipsticks. *“I heard that some of them went for the training…like six months ago, been* advised *not to follow the urine dip any more.”* Care home staff 7 ***(Knowledge)*** One stakeholder reported optimism that their work around UTIs, implementing the leaflet and decreasing dipstick use in their region had reduced the amount of urines being bought in to general practice. *“receptionist love me because I stop that wave of urine that used to come in* every *morning, and the nurses said it was taking hours of their time.”* Stakeholder 8 ***(Optimism)*** All general practice staff reported that they have had issues with patients bringing in urine samples to reception for dipping. *“lots of patients just dropping in samples that we never knew what they were for or whether to send it off, so we’ve tightened up on that.”* General practice staff 3 ***(Social influence)*** Care staff decided to use urine dipsticks as a result of noticing other symptoms. *“we usually notice something else which has caused us to do that test anyway…so we’re not just relying on that.”* Care home staff 2 Care home staff felt pressured by GP staff to use and report dipstick results for suspected UTIs. *“they’ll ask if you’ve done a urine dip, you’ll say, yeah, you’ll have to tell them what it’s showing.”* Care home staff 4 ***(Social influence)*** Some clinicians feel pressured by care homes to prescribe antibiotics based on a urine dipstick result. *“Sometimes we get a call from the care homes, they dip the urine and if it is positive and then they want antibiotic.”* General practice staff 2 ***(Social influence)*** Some care homes intended to keep using urine dipsticks to identify UTIs. *“Because it’s worked for us. It seems to have worked, I think that’s the hard thing,* *because it always has seemed to* work *that way.”* Care home staff 1 ***(Intentions)*** Some care homes intended to stop using urine dipsticks moving forwards. *“We feel that if it’s not required then it’s one less thing that you have to try* and *get from people.”* Care home staff 3 ***(Intentions)*** **Presentation of UTI** Many clinicians expressed that diagnosing UTI in older adults can be very difficult. *“often with UTIs, especially in old people…you’re not quite sure what’s going on…it might be a UTI…they’re just given a prescription with no one really* finding *out what’s going on, and it’s a nightmare.”* GP 2 ***(Skills)*** Some care staff identified that other conditions can present like a UTI. *“Some of them will present as if it’s a UTI but it’s actually constipation.”* Care home staff 3 ***(Knowledge)*** Many care home staff expressed that residents will not or are unable to tell them about their symptoms. *“A lot of them either don’t recognise the symptoms or if you ask them they’re going to say yes anyway.”* Care home staff 7 ***(Social influence)*** General practice staff stated that care staff sometimes provide vague information. *“they say the patient looks a little bit more confused today or a little bit more agitated, it’s not unusual, some of the behaviour, but again, that’s again vague.”* General practice staff 2 ***(Social influence)*** One GP stated that they were mindful that atrophic vaginitis can cause urinary symptoms and present like a UTI. *“they’ve had tummy pain, dysuria, frequency and it’s cloudy and they haven’t got any itching, then I would treat it as a UTI but…especially in older women, I’m always thinking about have they got atrophic vaginitis, especially if it’s a recurrent thing.”* GP 2 ***(Memory, attention and decision processes)*** **Urine samples for culture** Care staff expressed difficulty in obtaining urine samples, especially if the patient is incontinent or has dementia. *“you try and get a urine sample* but *that’s normally fairly tricky because either they use continence aids…Or they’ll go to the toilet and then you’ll have faeces with the sample.”* Care home staff 3 ***(Skills)*** A few GPs used urine culture results as a diagnostic tool. *“I’m not going to start antibiotics until I have obvious MSU showing there is an infection or not.”* General practice staff 2 ***(Knowledge)*** **Facilitators** Some practices had developed their own diagnostic template to aid UTI diagnosis. *“So we developed this* system *on protocol which we’ve not used before, for clinical staff to use like a prompt and help decision making processes.”* General practice staff 1 ***(Environmental context and resources)*** All care staff were confident in their ability to identify early signs of illness. *“We’re fairly observant of the symptoms and quite good at noticing* changes *in people and when they might be unwell.”* Care home staff 2 ***(Beliefs about capabilities)***	**Hydration and drinking** All older adults knew that hydration could prevent or help manage a UTI. *“I’ve been drinking water since it’s coming out my ears. Yeah, I’ve been trying to drink as much fluid as I can, so.”* Older adult 2 ***(Knowledge)*** All care homes actively encouraged residents to keep hydrated. *“I would say actually physically passing the drink to them, so you would encourage them* to *drink and usually they say, oh you know, I’ve had a lot today. We say, oh well just a little bit more and try and just sort of encourage them.”* Care home staff 3 ***(Skills)*** All general practice staff encouraged hydration as a preventative and self-care method. *“Hydration is what I focus on.”* GP 1 ***(Skills)*** Some care homes would decide to encourage drinking before concluding that the resident has a UTI. *“as harsh as it sounds we give them a drink and see if that perks them up and we see how far the confusion goes, we don’t automatically think UTI, it could be dehydration.”* Care home staff 4 ***(Memory, attention and decision processes)*** One resident described drinking less in order to avoid urinating at night. *“because I keep going at night. Which isn’t right…I’m not drinking more. I* hopefully *am drinking less.”* Care home resident 2 ***(Goals)*** Care staff believed that residents do not want to drink to avoid visiting the toilet regularly. *“they get worried about drinking too much because they don’t want to keep going to the toilet.”* Care home staff 7 ***(Social influence)*** **Antibiotics** Many general practice staff were prescribing UTI antibiotics over the phone to care home residents. *“in the volume of work it’s often, as you quite* rightly *say it’s often over the phone.”* GP 1 ***(Environmental context and resources)*** A few general practice staff reported prescribing antibiotics for UTI as a result of demanding patients. *“there is always still that pressure to prescribe. I came here because I’ve got a urine infection and you are going to prescribe me antibiotics no matter what you think.”* Nurse practitioner 3 ***(Social influence)*** Older adults do not mind taking antibiotics as long as it makes them well. *“I just want to feel well, and I don’t care what I take to feel like me you know.”* Care home residents 1 ***(Goals)*** Older adults aware of D-mannose were receptive to trying it as an antibiotic alternative. *“I went in and she immediately said I’ve been looking* something *up for you and she’d found them, they’re expensive but if it’s going to work then I’ll pay the money.”* Older adult 3 ***(Social influence)*** A few general practice staff expressed interest in conducting a UTI antibiotic audit. *“Auditing the antibiotic use would be really interesting to do, if* we *could do that that would be good.”* General practice staff 1 ***(Intentions)*** One general practice mentioned auditing their UTI antibiotics. *“We’ve re-audited the antibiotic prescribing…it’s kind of improved…my* trimethoprim *prescribing’s halved.”* General practice staff 3 ***(Behavioural regulation)*** **Perceived role in UTI management** Care staff and general practice staff were confident in their ability to manage diagnosed UTI. ***(Beliefs about capabilities)*** All care staff reported changing soiled incontinence pads immediately, even if the resident has a limited pad allowance. *“So, the residents are restricted on how many day or night pads that they’re assessed or allocated but if we find a resident that is soiled or their pad is wet we automatically change it.” Care* home staff 5 ***(Environmental context and resources)***

## Data Availability

The anonymised data presented in this study are available on request from the corresponding author. The data are not publicly available due to the sensitive nature of the topic.

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
