# Peer review of "A Qualitative Investigation of the Acceptability and Feasibility of a Urinary Tract Infection Patient Information Leaflet for Older Adults and Their Carers"

_antibiotics, 2021, doi:10.3390/antibiotics10010083_

Round 1

Reviewer 1 Report

Comments and Suggestions for Authors

I want to thank you again for the opportunity to write a review.  The introduction is well written. The background of the problem and the goal of the research are well presented. Only a small change is needed in the terms of formatting the text. Also, it is necessary to correct several grammatical and spelling mistakes.

First of all, the Materials and methods section must be placed within the text before the Results section.

Line 27: it appears that life threatening is missing a hyphen.

life-threatening

Line 34:

bloodstream, not blood stream

Line 48: it appears that evidence based is missing a hyphen.

evidence-based

Line 54: please use past tense for the sentence.

The aim of this study is to:

The aim of this study was to:

Line 79: Title 2.2. Key findings. The article is incorrectly numbered because I do not see article 2.1. within the text of the results. I think it is necessary to correct the numbering of articles.

Line 99: The noun phrase provision seems to be missing a determiner befor it. Consider adding an article.

the provision

Line 113: The noun phrase use seems to be missing a determiner before it. Consider adding an article.

the use of less...

Line 136: the amount

It appears that the quantifier the amount does not fit with the countable noun urines. Consider changing the quantifier of the noun.

This is my suggestion: ...leaflet use and diagnostic guidance had reduced the number of unrequested urines...

Consider adding an article or determiner in the next lines.

Line 152: the combined use...

Line 165: the feasibility...

Line 181:

a provision

Line 185:

...the inclusion of three-day antibiotic courses, the inclusion of D-mannose...

Line 216 and 271:

the provision

Line 197:

real-world

Line 200-201:

... drawing comparisons across different leaflets, audiences and conditions were inherently difficult.

Line 205: Consider changing the following nouns to the plural form.

... references

Line 269:

... the leaflets

Line 270: Check the spelling.

... behaviour

Line 279: The word seems to be miswritten.

... widespread

Line 281:

A greater or The greater

Line 311-312:

... were taken into account

Line 317:

follow-up

Line 348:

... pilot data were included

Line 369:

audio-recorded

Line 370:

data were stored

Author Response

I would like to thank each of the reviewers for providing their thorough and detailed feedback. I have read each comment and have provided a response to each comment below in blue. I believe that the changes made based on the reviewer comments has improved the paper markedly and I am very grateful for their thoughts.

Reviewer 1

I want to thank you again for the opportunity to write a review.  The introduction is well written. The background of the problem and the goal of the research are well presented. Only a small change is needed in the terms of formatting the text. Also, it is necessary to correct several grammatical and spelling mistakes.

Thank you for reviewing our paper and providing constructive feedback.

First of all, the Materials and methods section must be placed within the text before the Results section.

The order of the manuscript is in accordance with the journal requirements.

Line 27: it appears that life threatening is missing a hyphen. life-threatening

Thank you for identifying these grammatical and spelling mistakes. I have amended this in line with your suggestion

Line 34: bloodstream, not blood stream

Agreed, amended.

Line 48: it appears that evidence based is missing a hyphen. evidence-based

Agreed, amended.

Line 54: please use past tense for the sentence. The aim of this study is to: The aim of this study was to:

Agreed, amended.

Line 79: Title 2.2. Key findings. The article is incorrectly numbered because I do not see article 2.1. within the text of the results. I think it is necessary to correct the numbering of articles.

Thanks for identifying this, I’ve now amended the numbering.

Line 99: The noun phrase provision seems to be missing a determiner before it. Consider adding an article. the provision

Agreed, amended.

Line 113: The noun phrase use seems to be missing a determiner before it. Consider adding an article. the use of less...

Agreed, amended.

Line 136: the amount. It appears that the quantifier the amount does not fit with the countable noun urines. Consider changing the quantifier of the noun. This is my suggestion: ...leaflet use and diagnostic guidance had reduced the number of unrequested urines...

I agree, I have added your suggestion.

Consider adding an article or determiner in the next lines.

Line 152: the combined use...

Agreed, amended.

Line 165: the feasibility...

Agreed, amended.    

Line 181: a provision

Agreed, amended.

Line 185: ...the inclusion of three-day antibiotic courses, the inclusion of D-mannose...

Agreed, amended.

 Line 216 and 271: the provision

Agreed, amended.

Line 197: real-world

Agreed, amended.

Line 200-201: ... drawing comparisons across different leaflets, audiences and conditions were inherently difficult.

Agreed, amended.

Line 205: Consider changing the following nouns to the plural form.... references

Agreed, amended.

Line 269: ... the leaflets

Agreed, amended.

 Line 270: Check the spelling.... behaviour

Agreed, amended.

Line 279: The word seems to be miswritten.... widespread

Agreed, amended.

Line 281: A greater or The greater

Agreed, amended to ‘A greater…’.

Line 311-312: ... were taken into account

Agreed, amended.

Line 317: follow-up

Agreed, amended.

Line 348: ... pilot data were included

Agreed, amended.

Line 369: audio-recorded

Agreed, amended.

Line 370:data were stored

Agreed, amended.

Reviewer 2 Report

The authors have submitted a manuscript of interest for its evaluation and possible publication. However, the text presented presents a series of points that should be modified before a possible publication. The reviewer will describe the points following the order of the sections of the manuscript. Abstract section The reviewer advises the authors to describe a brief background of the situation, the objectives of the study and some minimal conclusions.

Methods section
The reviewer does not understand why the authors do not place the methods section after the introduction and before results, since this is the usual order in a scientific text. Also in the methods section, the reviewer advises the authors to describe the eligibility criteria, the calculation of the sample size and the power of the study. Lastly, the reviewer advises the authors to describe whether the qualitative study was cross-sectional or longitudinal.
RResults section The reviewer suggests that the authors order the results so that the reading gains understanding.
Discussion section the reviewer advises the authors to reduce the length of this section. Likewise, the reviewer considers that the authors should describe the limitations of the study in this section and that, after the limitations, the authors place the conclusions. Also, the reviewer has doubts about the "detailed insight into the acceptability and feasibility of using the UTI leaflet for older adults ... "taking into account the sample size. For this reason, the reviewer advises the authors to rewrite the conclusions.

Author Response

I would like to thank each of the reviewers for providing their thorough and detailed feedback. I have read each comment and have provided a response to each comment below in blue. I believe that the changes made based on the reviewer comments has improved the paper markedly and I am very grateful for their thoughts.

Reviewer 2

The authors have submitted a manuscript of interest for its evaluation and possible publication. However, the text presented presents a series of points that should be modified before a possible publication. The reviewer will describe the points following the order of the sections of the manuscript.

Thank you for taking the time to review our paper and provide constructive feedback.

Abstract section The reviewer advises the authors to describe a brief background of the situation, the objectives of the study and some minimal conclusions.

I have now included sentences at the start of the abstract detailing background and study objectives. I have also highlighted the recommendations at the end of the abstract which serve as a conclusion to the study.

Methods section
The reviewer does not understand why the authors do not place the methods section after the introduction and before results, since this is the usual order in a scientific text.

This was also pointed out by reviewer 1. This layout is in accordance with the journal requirements.

Also in the methods section, the reviewer advises the authors to describe the eligibility criteria, the calculation of the sample size and the power of the study.

In section 4.1 participant selection is detailed including rationale for GP practice selection and care home eligibility based on inspection ratings. In order to highlight this for readers I have changed the title of this section to ‘4.1 Participant selection and eligibility’.

As this is a qualitative study we did not feel it was appropriate to calculate sample size or the power of the study. We believe these quality criteria are more suited to quantitative studies.

Lastly, the reviewer advises the authors to describe whether the qualitative study was cross-sectional or longitudinal.

Thank you for pointing this out, I’ve now added it into the design section of the methods.

Results section. The reviewer suggests that the authors order the results so that the reading gains understanding.

In response to this and other reviewers comments regarding the presentation of the results. I have now re-ordered Table 1. I have re-structured the table to be sectioned into topics rather than the TDF domains. The domains are still reported after each finding but I have now grouped each finding by their relevant topics which should make it easier for the reader to follow.

Discussion section the reviewer advises the authors to reduce the length of this section.

Based on the comments from other reviewers to include discussion aspects felt to be missing I have not reduced the length. However, I am open to reducing the length of the discussion if this is required to go for another round of reviews, and I am open to suggestions as to what to remove.

Likewise, the reviewer considers that the authors should describe the limitations of the study in this section and that, after the limitations, the authors place the conclusions.

I appreciate that your suggestion is usual practice, however, in line with the journal requirements, I have placed the conclusions at the end of the paper, following the methods section. I have however provided an implications section following the strengths and limitations.

Also, the reviewer has doubts about the "detailed insight into the acceptability and feasibility of using the UTI leaflet for older adults ... "taking into account the sample size. For this reason, the reviewer advises the authors to rewrite the conclusions.

Thank you for your comment. I have reworded the opening sentence to the conclusions in line with your suggestion to ‘This novel study has provided insights into the acceptability…’.

Reviewer 3 Report

Mostly minor comments.

Line 53: Is there a section about AMR in Appendix A?

Lines 61-64: Relocate to Materials and Methods.

Would be helpful to detail Section 4 Materials and Methods before Results and Discussion.

Re recruitment: Of the 304 that received a invitation letter, 291 did not participate. Does this mean all these care facilities were considered inadequate or refused?

Appendix B: For conciseness, remove 'notes' rows

Author Response

I would like to thank each of the reviewers for providing their thorough and detailed feedback. I have read each comment and have provided a response to each comment below in blue. I believe that the changes made based on the reviewer comments has improved the paper markedly and I am very grateful for their thoughts.

Reviewer 3

Mostly minor comments.

Line 53: Is there a section about AMR in Appendix A?

Yes, in the centre section bottom right there is a section called ‘always trust your pharmacists’/nurses’/doctors’ advice about antibiotics’, and it explains briefly how resistance works along with side effects and the importance of preserving antibiotics. It is not explicitly labelled antimicrobial resistance as in the development stage we believed that the term antimicrobial resistance would not be familiar to many.

Lines 61-64: Relocate to Materials and Methods.

Would be helpful to detail Section 4 Materials and Methods before Results and Discussion.

Thank you. I appreciate that your suggestion is usual practice, but the order of the paper is in line with the journal requirements.

Re recruitment: Of the 304 that received a invitation letter, 291 did not participate. Does this mean all these care facilities were considered inadequate or refused?

Thank you for identifying this. I have now clarified in the participant selection and eligibility section that even though all facilities received an invitation letter, very few responded expressing an interest in the study, hence the follow up phone calls. I also added that only 3 care homes were excluded due to their ‘inadequate’ rating by the CQC.

Appendix B: For conciseness, remove 'notes' rows

Agreed, I have now removed the ‘notes’ sections and this has saved space.

Author Response

I would like to thank each of the reviewers for providing their thorough and detailed feedback. I have read each comment and have provided a response to each comment below in blue. I believe that the changes made based on the reviewer comments has improved the paper markedly and I am very grateful for their thoughts.

Reviewer 4 (I have taken the comments in order from the PDF and inserted below for ease of reference)

how was the leaflet developed? did the TDF assist with the development of the leaflet and/or the interview questions?

This is a good point, thank you for highlighting this. I have now added a section into the introduction explaining that the TDF was used in the development phase to inform the interview questions.

Not clear in the manuscript whether general practice staff are GPs and/or PNs?

Agreed. I have now added a section into the abstract and results explaining that general practice staff are made up of GPs, nurses and health care assistants.

definition? for those not in the UK system. Do care staff include nurses and physicians?

Agreed. I have now added a section into the abstract explaining that care staff were made up of nurses and carers. This is also further elaborated on in the results section.

I really like this table, however, it might be a bit much in the text, as this is very detailed and a lot of information - suggest putting it in the appendix and have a summary of the significant/highlights from these participants, ie. age, years of experience etc

Agreed, the figure is quite text heavy. I have now moved this to an Appendix and have referred readers to the appendix for further recruitment details. The details collected about participants have been referred to in the text, but we did not collect data on age and years of experience.

how will you address this? (Referring to: Most older adults would be happy to receive the leaflet, although some concern was raised by clinicians as to the leaflet’s acceptability for older adults who are coping with multiple health issues and may find the information overwhelming.)

This is a very interesting point. I have addressed this concern in lines 277-280 by explaining that where residents lack capacity or may find the leaflet overwhelming, the leaflet could be given to family and friends of residents to provide education and to reinforce health behaviour messages from staff around hydration, self-care and prevention.

was it (the leaflet) difficult for patients to understand?

This is a very good point. I have highlighted in the strengths and weaknesses section that only older adults with full capacity could take part in the study, therefore we didn’t speak to any older adults who had difficulties in understanding the content. I explain that this is a limitation as no data was captured for those older adults deemed unsuitable to receive the leaflet due to the content being ‘overwhelming’.

check re. promotion, dissemination of the leaflets, and how they recruited the participants: Was the recruitment wide? did they reach data saturation? Did they have a goal of how many to recruit?

is there any way to measure whether they have used it?  Did all your participants who used it and found it helpful?

In the leaflet implementation section, I have added a statement to explain that each region aimed to saturate their area with the leaflet, but due to lack of uptake monitoring neither region could be certain of the uptake.

In section 2.2.2 value of the leaflet, it is explained that all participants valued the leaflet and found it useful, but some had further suggestions for improvement which are detailed.

Unclear re. clinicians - how many in care homes and how many in general practice?

Thank you for pointing this out. I had used the term clinicians to refer specifically to general practice staff. I have now checked each mention of ‘clinicians’ and changed to ‘GP staff’ where appropriate to add clarification to readers.

Did you conduct the focus groups with people with different roles?

In the data collection section I have added in a statement to clarify that the focus groups were heterogenous, in that all roles were permitted to attend the focus groups.

what is the working relationships between care homes and general practice?  Not clear.

In the barriers and facilitators section I have clarified that this finding is specific to this study and that working relationships between care homes and general practice will vary across regions and depending on the individual facility.

how are general practices related to care homes?  Do they go into the homes and visit patients?  How/when do they use these information sheets?

Is there a patient age limit?

I’ve added into the introduction that clinicians can use the leaflet with older adults in the community and in care. I have also added into the results that the general practices services both care homes and the community. During the discussions, the distinction was not always made as to whether the patients referred to were patients in the community or care homes, but where a distinction was made this has been highlighted in the results and discussion.

There is no patient age limit and I have now added this into the discussion section to explain that the leaflet can be used with older adults with no upper age limit, but can also be shared with carers or family members if necessary.

is this from the care staff's point of view of the older adults?  A quote here would support your findings.

I think this is a really important point and I have now added clarification that this was reported by both older adults and care staff, I have also added a quote to illustrate that one care home resident deliberately drank less.

Not sure how this relates to the patient information sheet - do they want more information about this? Should there be a leaflet for care staff since this is aimed at patients and carers?  What is the definition of care staff?  Are they trained care home staff?

This is a really important point. I have now clarified that care staff could identify when a resident is ill but had difficulty in distinguishing a UTI from other illnesses due to similar presentations.

I have also added to the introduction that the leaflet is designed for both community and care home carers.

I have also highlighted in the results that all of the carers in this study are care home carers.

or clinicians - this is not really related as the sheets are not aimed at them.

I have added a statement to this section explaining that clinicians also want additional resource and information around diagnosing UTI in dementia patients. This is also addressed in the discussion summary as something that cannot easily be solved by a leaflet and will require additional resource.

so this is a problem in dissemination of the leaflet.

Agreed, this is discussed as a limitation in the discussion along with possible reasons for why this is the case.

what about the patients?  Did they find it difficult to use and understand the leaflets? or their carers?

This was raised in a previous comment therefore I have copied my response to offer an explanation - This is a very good point. I have highlighted in the strengths and weaknesses section that only older adults with full capacity could take part in the study, therefore we didn’t speak to any older adults who had difficulties in understanding the content. I explain that this is a limitation as no data was captured for those older adults deemed unsuitable to receive the leaflet due to the content being ‘overwhelming’.

results seem staggered, it would be better to somehow linked together, to compare an contrast the different roles

In response to this and other reviewers comments regarding the presentation of the results. I have now re-ordered Table 1. I have re-structured the table to be sectioned into topics rather than the TDF domains. The domains are still reported after each finding but I have now grouped each finding by their relevant topics which should make it easier for the reader to follow and easier to compare the various roles based on each topic.

what is the role of this stakeholder?  How did they reduce the amount of urines (or number of urine samples?) being brought into general practice?

Thank you for pointing this out. I have now added to this section of the table that this stakeholder had worked on UTIs by implementing the leaflet and ultimately reducing dipstick use in their region.

How does this relate to the leaflet?

In the results section 2.1.4 it is explained that atrophic vaginitis can sometimes present like a UTI and should be considered as an alternative cause to urinary symptoms. Therefore, in the summary section in the discussion it is discussed how atrophic vaginitis will be added into the leaflet as something which can present like a UTI. To add clarification to the table I’ve added that this can cause urinary symptoms and therefore present like a UTI.

There seems to be a lack of discussion on some of the issues identified from the findings in the TDF Did your results answered the objectives?

In response to this and other reviewers comments regarding the presentation of the results. I have now re-ordered Table 1. I have re-structured the table to be sectioned into topics rather than the TDF domains. The domains are still reported after each finding but I have now grouped each finding by their relevant topics which should make it easier for the reader to follow. This has also meant condensing the findings to only the relevant topics.

how will this leaflet assist in the change in behaviour?

I’ve highlighted in the conclusion an overview of the behavioural changes, but also how the recommended changes are needed in order to see further behavioural changes.

what is in the information form?

I have now clarified that this is the study information form to inform participants about their involvement in the study, rather than an information form around UTIs.

clever, but check how this turned out - did they have a even response from participants with high prescribing. Not clear how they selected those regions

In the section for participant selection and eligibility I have added a statement to explain that the two regions were selected because of their intention to implement the leaflet in their areas and their willingness to support the study. I have also clarified that prescriptions at an individual facility level was not monitored.

define consumption - is that the same as prescribing? - or is this different because they are from the care homes?

Agreed, thank you for pointing this out. I have now changed this to ‘antibiotic prescribing at a primary care level’.

selection bias?

This is very true and is discussed in the strengths and limitations as a limitation to the study that only older adults with full capacity and understanding were chosen to participate.

experience in TDF?

Agreed, I have now added this into the researcher context section.

Round 2

Reviewer 2 Report

The authors have made an important effort to improve the manuscript Nevertheless